# IoT Data Sharing Platform in Web 3.0 Using Blockchain Technology

Abdul Razzaq [1,*], Ahmed B. Altamimi [2,*], Abdulrahman Alreshidi [3], Shahbaz Ahmed Khan Ghayyur [4], Wilayat Khan [5] and Mohammad Alsaffar [3]

1   Ocean Technology and Engineering, Ocean College, Zhejiang University, Zhoushan 316021, China
2   Department of Computer Engineering, University of Ha'il, Ha'il 81481, Saudi Arabia
3   Department of Information and Computer Science, University of Ha'il, Ha'il 81481, Saudi Arabia
4   Department of Computer Science and Software Engineering, International Islamic University, Islamabad 44000, Pakistan
5   Department of Electrical and Computer Engineering, COMSATS University Islamabad, Wah Cantt 47010, Pakistan
*   Correspondence: 11934071@zju.edu.cn (A.R.); altamimi.a@uoh.edu.sa (A.B.A.)

**Abstract:** As Internet of Things (IoT)-based systems become more prevalent in the era of data-driven intelligence, they are prone to some unprecedented challenges in terms of data security and systems scalability in an era of context-sensitive data. The current advances in IoT-driven data sensing and sharing rely on third-party sources of information (TTPs) that gather data from one party, then transmit it to the other. As a result of TTPs' involvement, such IoT systems suffer from many issues including but not limited to security, transparency, trust, and immutability as a result of the involvement of the company. Moreover, a multitude of technical impediments, such as the computation and storage poverty of IoTs, privacy concerns, and energy efficiency, enhances the challenges for IoTs. To address these issues of IoT security, we propose a blockchain-enabled open IoT data-sharing framework based on the potential of the interplanetary file system (IPFS). We have used a case study-based approach to evaluate the proposed solution. It is submitted that the proposed scenario is implemented by building smart contracts in Solidity and deploying them on the local Ethereum test network, using the Solidity programming language. With the implementation of smart contracts on the blockchain for access roles in IoT data sensing, the proposed solution advocates for a blockchain-based approach to data security for IoT systems that makes use of smart contracts for access roles.

**Keywords:** Web 3.0; Internet of Things; blockchain; smart contract; distributed storage; IPFS; data sharing

## 1. Introduction

In an Internet of Things (IoT), physical objects (things) that are connected to the internet are equipped with software and sensors which allow them to exchange data with the rest of the world's systems and devices by means of the internet. The recommended method for secure data sharing fails in these circumstances because of the volume of data created, different devices [1], lack of confidence as well as participants, and the lack of openness in data management.

The information interconnection of the entire production process is the key component of Industry 4.0. In order to transition industry production to the industry 4.0 age, information physics system development must be accelerated. In each stage of production, businesses use a sizable number of sensors and actuators, but each one can only affect the subsystem to which it belongs. The Internet of Things' effectiveness is constrained by the close coupling of components [2,3].

The first version of the internet, known as Web 1.0, denotes the beginning of the internet in the late 1980s. Only static "read-only" messages created by a small number of users were included. The development of web 2.0, which placed a strong emphasis on enhancing user engagement and interaction, was then observed around the world. Users were able to create accounts using a variety of Web 2.0 apps, allowing them to establish distinctive online personas. With the advent of web 3.0, the globe is now moving toward the most recent paradigm in the web's evolution. What benefits does the new internet have to offer, then? Let us learn more about the new way of looking at the internet and the technologies which will be key in igniting this new revolution.

In essence, interconnection and interoperability are two key qualities that the IoT inevitably requires [4]. While interoperability refers to how IoT devices may swap information and utilize that information to carry out data analytics [5], interconnection refers to how these devices are connected to one another through ubiquitous networks with high-speed transmission. In other words, to break the data isolation in IoTs, which also supports decision-making capability at the system level [6,7] for an extensive range of industrial architectures, seamless data exchange among multiple industrial sectors and their systems, such as carriers, suppliers, and manufacturers, is necessary. For instance, logistics companies can optimize the schedule of delivering packages in order to cut down the delivery time [8] and significantly reduce delivery costs when they receive road conditions and provided real-time traffic information from the industry.

Blockchain technology has emerged as a potential solution in several distributed applications where trust and transparency are essential aspects. As a result, it is not unexpected that both businesses and academia are debating how to effectively merge IoT systems with blockchains. To address the issue of secure data interchange, a number of research projects suggest directly connecting IoT systems to a blockchain platform [9,10]. The vast majority use hybrid storage strategies like a provider who keeps the data current, while the blockchain offers benefits such as integrity and reliable distribution [11]. Authors propose, for example, storing access control strategies that the storage provider queries as it receives an access request. As a result, the storage provider operates as a hub for making and enforcing policy decisions, while the blockchain safeguards policy integrity and enables open audits of policy changes.

The research community has made technical advances in the past decade to support data-sharing methodologies. Collaboration and wise judgments can help research-based activities develop in this way. Data sharing is a necessary step in maximizing the benefits of scientific advances [12]. However, it is critical to understand when the best moment is to share the data. Before beginning the data exchange procedure, these questions must be answered completely. By employing the resources of blockchains [13], this research allows for protected data sharing and sale. In the realm of information technology, blockchain, or a distributed ledger, is a novel trend. Blockchains have been used in several financial and non-financial applications.

The centralized authorities known as cloud servers store a vast amount of data [14]. A single-point failure is one of the potential hazards associated with a dominant authority. To avoid any catastrophe, data backup services from third parties are used. The issue is that network nodes have storage and processing constraints. A peer-to-peer framework named IPFS is being used for this purpose [15].

Among peer-to-peer protocols, IPFS is content-based, and assigns a cryptographic hash to each IoT data file. The hash is targeted to make the text unchangeable [16]. By cutting bandwidth costs, speeding up IoT data downloads, and sharing vast volumes of data without duplication, IPFS allows storage savings. Up to 256 KB of unstructured binary data can be stored in a single IPFS object. If the data is over 256 KB, it is split up and stored as IPFS objects with one empty object connecting the IoT data files. The IPFS storage system is therefore an immutable storage system since, if a file's hash value is modified, it will affect the hash value of the file. The IPFS data transport protocol supports hash string routes. Encrypted data and additional information can be stored in it.

The system architecture design is shown in Figure 1, which is intended to guide system developers in maintaining the layer of abstraction that is maintained throughout the system development process. As a result, there are three layers which are all interconnected. The first layer consists of a deployed sensor system, where all of the sensors are deployed and all of them produce data in package form. The second layer is a data processing algorithm, and the third layer is a data analytics system that provides readable data to be analyzed. During the cryptographic process, the blockchain ledger receives data from the IoT data server, which stores all the data generated by the sensors. Some of the contributions that this study can provide include the following:

- Enable trust-based access management—implemented via smart contracts—to enable access control and authorization for IoT-based security-critical data.
- Modularize the solution with algorithmic implementation that automates and customizes the solution with parameterized input from the users.
- Validate the solution via a scenario-driven approach to assess system performance based on algorithmic execution and query response times.

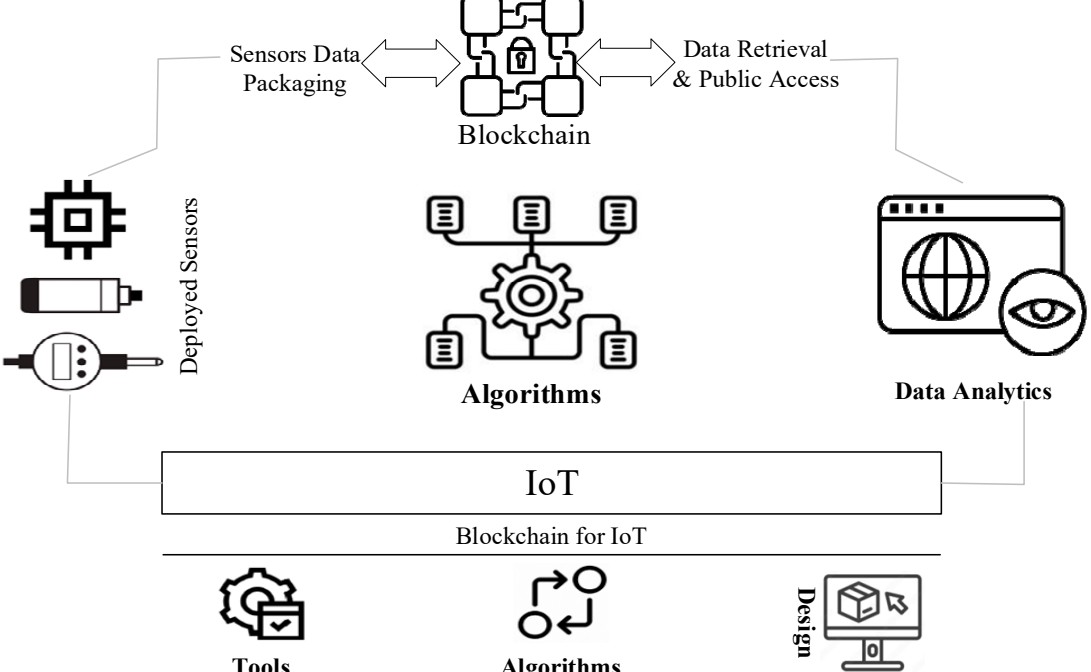

**Figure 1.** Overview of the recommended Model.

*Paper Organization*

As for the remainder of the paper: Section 2 presents the state-of-the-art, while Section 3 presents the rationale and problem description. In Section 4, we examined the algorithmic design, technology implementation, and system model of the proposed scheme and smart contracts. Section 5 contains details about the evaluation and simulation results. The last section, Section 6, concludes the paper.

## 2. Existing Work and Technical Challenges

This section provides background information to assist in putting the components of a blockchain-based IoT data-sharing system into perspective. Additionally, we review and impartially contrast the most pertinent studies that are currently available in order to support the contributions and scope of the proposed model.

Despite its promising qualities, a security problem [17] will always exist which prevents open data sharing in the IoT. After shared information has been received by numerous recipients, the data owner has little control over who can view the information. In most

data-sharing situations, the sender merely permits the recipient to make use of the provided data, and does not let the recipient divulge the shared data to other parties or the general public without authorization for the goal of profit or other self-interest. It is essential that if there is a data leakage incident, the sending party should be found and held accountable, regardless of whether the data leakage occurred intentionally or accidentally (for example, if the sending party is aware of the data breach and had obtained the leaked data through the Internet).

For auditable private data sharing, Kokoris-Kogias et al. [18] introduced CALYPSO, where access control laws are enacted, and data is stored on-chain by a collective authority made possible by the blockchain. The massive amounts of IoT data generated by numerous IoT devices in real-world systems are too much for this method to manage. A system for exchanging time-series IoT data was developed by Shafagh et al., and it requires data owners to make transactions in order to set policies each time the data is shared with a new party. After that, only the proprietor is allowed to make changes to the policy [19].

To get the most out of the research's capabilities, data exchange is essential. The literature proposes and discusses a variety of data exchange strategies. There is not enough research on incentive mechanisms to encourage data sharing. To address these flaws, the authors of [20] performed a study of health and medical data in order to find incentive processes and compare pre- and post-empirical outcomes. According to the survey, the rate of data sharing for a single reward for medical and health data is being analyzed. As a result, it is argued that further incentive-based research is required to stimulate data collection.

The Internet of Things significantly enables in the automation of our everyday lives (IoT). Information is frequently shared and exchanged between electronic devices online [21]. A system must be created to ensure data integrity and digital device authentication due to security and privacy concerns. The authors of [22] proposed a decentralized blockchain-based scenario called a "bubble of trust." However, the suggested technique has certain drawbacks, such as the inability to adjust to a real-time setup, the need for an initiation step, and the lack of discussion of cryptocurrency rate progression.

The blockchain-based IoT data-sharing schemes have drawbacks, including security concerns, high maintenance costs, and the monitoring of enormous amounts of data coming from IoT networks [23]. The output of smart industries depends on data collected from IoT devices or their DTs. The data that is gathered may come from erroneous sensors, RFID, actuators, or their DTs, which introduce inaccurate data for analysis and action [24,25]. The authors proposed a secure fabric-based data transport system as a solution to these problems. Data is stored using a data consensus technique through a dynamic linked-assisted storage system. But power data security is neglected, and this technology is only suggested for modest uses [26].

The blockchain has a substantial storage problem, particularly when large volumes of data must be retained on network nodes. Because it does not support the storage of very large files, terminal node storage capacity is constrained. This conundrum leads to several problems, such as the need for great computing power and the high computational cost for vast amounts of data. In response to these problems, Stiechen et al. [27] presented an IPFS-based decentralized storage technique. The files are segmented on each node; on the other hand, until users are granted the proper rights, a file cannot be seen. This is a clever tactic for protecting sensitive information. The suggested schemes encounter latency when downloading files from the server because of blockchain interaction, and they do not provide real-time data saving.

Table 1 lists the benefits and drawbacks of centralized and decentralized identity management systems. To demonstrate the differences between current blockchain-based systems and old central systems, we provided four major aspects.

**Table 1.** Comparison between centralized and decentralized.

| Acreage | | Conventional Systems | Blockchain Systems |
|---|---|---|---|
| 1- | Control | Centralized | Decentralized |
| 2- | Identity Change | Simple to alter details on the server. | History is unchangeable and secure to alter. |
| 3- | Storage | Centralized servers | Distributed Nodes. |
| 4- | Freedom | Identity theft is a possibility for users. | Ownership of the data is returned to the users. |

### 3. Research Methodology and Motivational Consequence of Solution

We now present the research methodology that gives the details of the design for a proposed solution. An overview of the research method is presented in Figure 2 which is based on four steps, following an incremental mechanism to analyze, design, implement, and validate the solution, as detailed below.

Figure 2 is the visualized overview of our research methodology and is divided into four different modules. In the first module of literature analysis, we conducted a critical analysis of the available literature of published research including a road map of technology and technical reports. We followed the recommendations to perform the literature analysis [28]. Prior to implementation, the solution is discussed in the second module of design. The third module of implementation has a thorough discussion of how the answer is implemented using computational and storage-intensive methods.

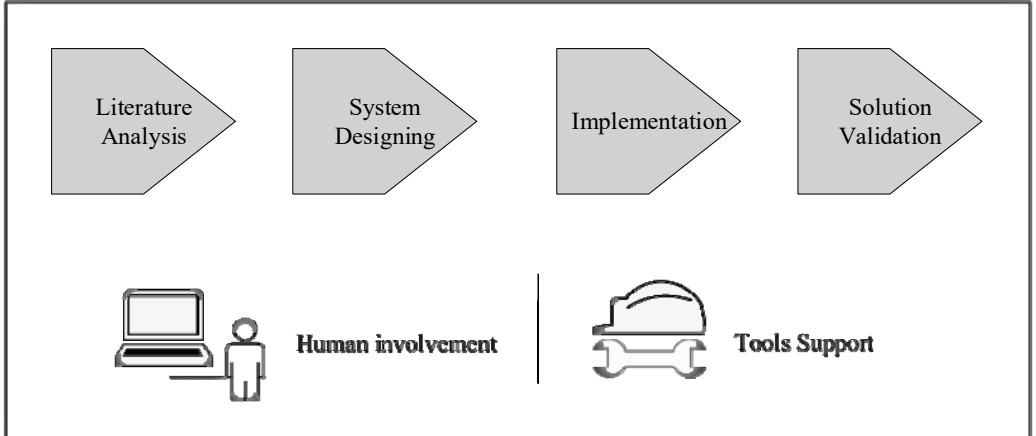

**Figure 2.** Illustration of Research Methodology.

The suggested system is summarized at an abstract level in Figure 3, where the module flow is shown. It is intended that all modules and stakeholders will communicate. Every component of the system design demonstrates the usage of data to illustrate the IoT idea. For instance, the data sensing module deals with gathering and representing data that is gathered from sensors, transmitted to the server, and stored in the database. System design helps programmers create and improve systems while abstracting away some implementation specifics that can be supplied with the right tools.

According to Figure 3, this system is composed of four layers: the sensing layer, the storage layer, the processing and blockchain layer, and the user layer. In addition to reading data from sensors, the sensing layer is responsible for packaging the sensing information for sending to the second layer of IoT storage for further processing. As part of layer 2, the data from all deployed sensors' data is stored in detail, the details of the sensing. The processing Blockchain layer 3 is used to save the transaction for each data-sharing action with the required detail. The fourth layer is the user interface layer used to share data.

We have obtained inspiration to work on digital data exchange utilizing blockchain based on the current research stated above. Although most researchers have worked in

comparable areas, there is a great room to improve and alter previous work in order to assist the research community.

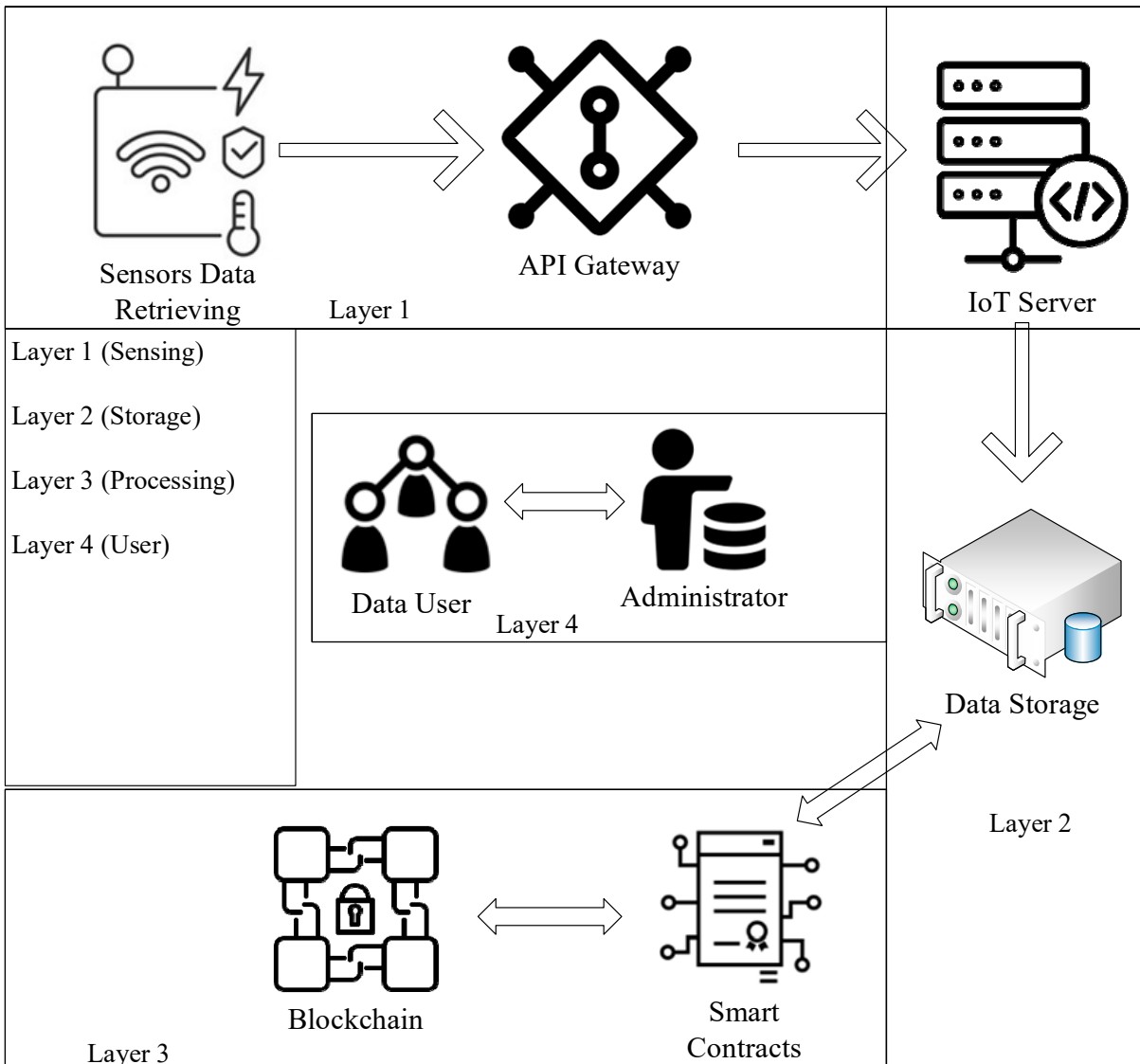

**Figure 3.** Overview of the proposed solution.

Figure 4 shows the flow of getting IoT data and storing it in a blockchain using a smart contract as the method for storing IoT data. The package file of IoT data is uploaded and saved the transaction record by the hash key when uploaded to IPFS. It is transferred to the DApp and uploaded. There are two kinds of uploading classifications in the DApp: one is carried out manually by the admin, and the other is carried out instantly by the system. The admin gets the hash key from the blockchain along with other necessary data and manually uploads the available package file of IoT data to IPFS. In other kinds, the system uploads immediately after receiving a fresh package file from the IoT server. Using the path, the system downloads the file from the IoT server, uploads it to IPFS, and then retrieves the back file hash that is recorded in the blockchain along with other information. Both execution processes are the same, but one is manual uploading by a user and the second is auto uploading by the system.

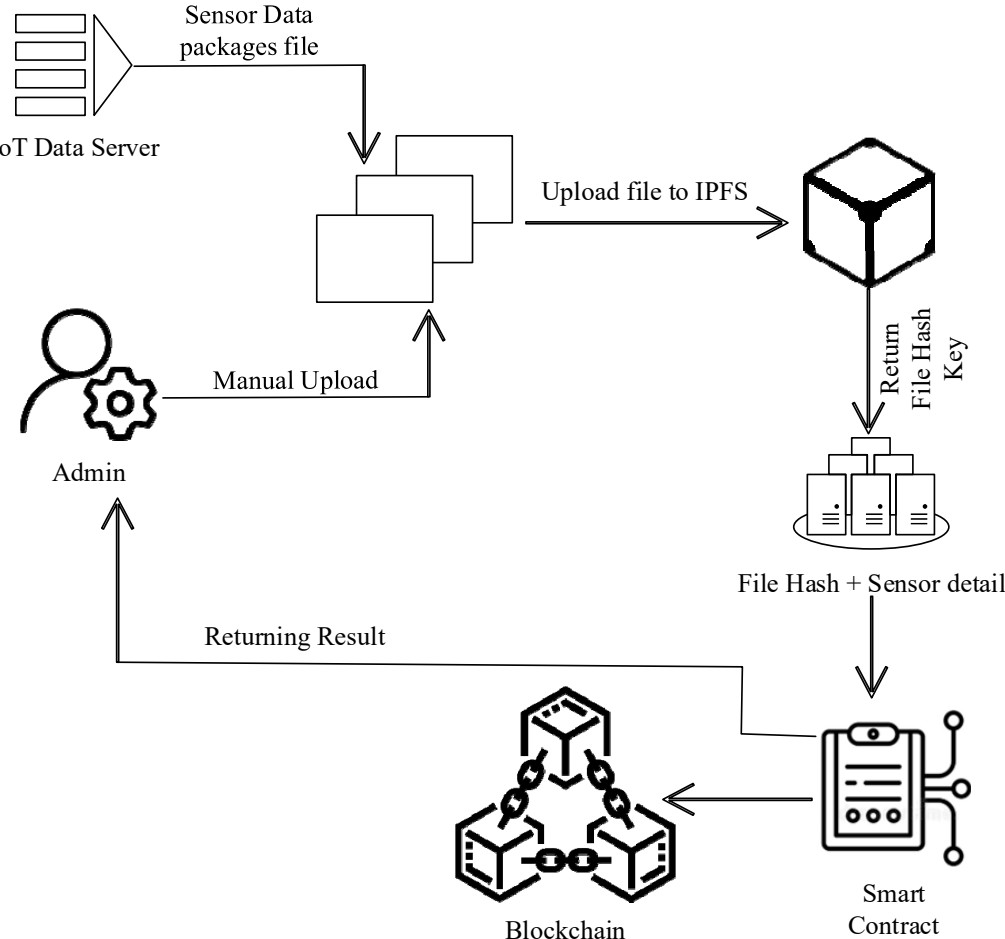

**Figure 4.** Data storing process.

The manual or system starts the digital data-sharing process by creating metadata for the original file. The metadata would contain details about the file's name, category, description, and size. Once finished, the information and the entire data file are uploaded to IPFS. Here is an illustration of a file transfer to IPFS (see in Listing 1):

**Listing 1.** Function source for uploading the data into IPFS and return hash.

```
//var UploadingType = ReactSession.get("uploadingType");
var func1 = performance.now();
console.log(func1);
var loc = document.getElementById("locationid"); //
locationid.options[locationid.selectedIndex].value;
var location = loc.options[loc.selectedIndex].text;
const sensor = this.sensorid.value;
const description = this.descriptionid.value;
ipfs.add(this.state.buffer, (error, result) => {
console.log('Ipfs result', result)
if(error) {
console.error(error)
return
}
sm1 = performance.now();
this.props.AddDataPackegeRecord(sensor, location, result [0].hash, description, 'Admin')
sm2 = performance.now();
})
```

When a file is uploaded to IPFS, it generates hashes of the contents and sends them back to the admin or system. When IPFS sends the hash to the admin or system, it maps the available parameters with the hash key. If this process is started by the admin (manually), admin will select the package file of IoT data and upload it manually to IPFS through the given system, and IPFS returns a hash key. The admin will map the required parameters (sensor, location, description) through the available input form and submit it to a smart contract where all data will be saved in the blockchain. The same execution process will be started for system uploading. The system will fetch the latest last uploaded file from the IoT data server by the given path; it will be uploaded by the system directly to IPFS, retrieve the hash key which will be mapped with available information, and stored in the blockchain through a smart contract. See the next code snippet (see in Listing 2):

**Listing 2.** Smart Contract Function to Record the Transaction in Blockchain Ledger.

```
function AddDataPackegeRecord(uint _sensorId, string memory _location, string memory
_hashKey,
string memory _desc, string memory _uploadingType) public{
dataUploadCount ++;
GetDataList[dataUploadCount] = DataUpload(dataUploadCount, _sensorId, _location,
_hashKey, _desc, _uploadingType, now);
GetData_sid[_sensorId] = DataUpload(dataUploadCount, _sensorId, _location, _hashKey,
_desc, _uploadingType, now);
GetData_date[now] = DataUpload(dataUploadCount, _sensorId, _location, _hashKey, _desc,
_uploadingType, now);
GetData_loc[_location] = DataUpload(dataUploadCount, _sensorId, _location, _hashKey,
_desc, _uploadingType, now);
GetData_sid_loc[_location][_sensorId] = DataUpload(dataUploadCount, _sensorId, _location,
_hashKey, _desc, _uploadingType, now);
emit DataUploadCreated(dataUploadCount, _sensorId, _location, _hashKey, _desc,
_uploadingType, now);
}
```

Phase 1 is a part of the sensors' data in IoT. There are also several sorts of sensor data. The data is packaged in a file for a certain time period, such as an IoT data package for 10 min, though it might be less or more. The gateway service sends this packet of IoT data to the IoT server. As an IoT server where all the data is processed of the deployed sensors, MSSQL is used in the same server to store the data.

Phase 2 is part of the system's service. There are two sorts of IoT data uploading categories in the DApp. Manually uploading and using a system, we have created a service called system service or auto uploading service for system uploading. The system service runs on the server's backend and makes a request to the IoT server to obtain the most up-to-date package file containing IoT data. The system service grabs the package file from the server and uploads it to an IPFS server, which then returns the hash of the file to the system. The smart contract performs the function to save the data in the blockchain by receiving the file hash and other necessary parameters from the system service. This service cycle of actions repeatedly occurs after a predetermined amount of time or is started by getting a package file of IoT data from the server.

Phase 3 is part of the manual uploading category by Admin; the technique for uploading a package file of IoT data to IPFS and storing it on the blockchain through a smart contract is the same, with the exception that this activity is conducted by the admin (manually).

Phase 4 is for accessing the existing data publicly. Users can view and download all IoT data packages for free from this open-access platform. By using a web portal, users can view the IoT data. The user will be able to access the data in a variety of ways, depending on their needs.

$$Failed_{\_Transactions} = \sum_{i=1}^{n} Total_{Requests} - \sum_{i=1}^{j} Accepted_{Requests} \qquad (1)$$

In order to determine the number of failed transactions, we take the number of accepted requests and subtract them from the total number of requests in the equation (i).

$$Successful\_{Transactions} = \sum_{i=1}^{n} Total_{Requests} - \sum_{i=1}^{j} Rejected_{Request} \tag{2}$$

Using Equation (2) and subtracting the number of rejected requests from the total number of requests, one can obtain the number of successful transactions.

## 4. Algorithms and Technologies for Solution Implementation

The specifics of the implementation are given in this section. A private network of the Ethereum blockchain makes up the proposed solution. Solidity is effectively used by the open-source distributed network Ethereum, a computer language that enables the creation of smart contracts.

### 4.1. Overview of System

- A lightweight cross-platform code editor called Visual Studio Code is available in the Microsoft Visual Studio Code product family. VSC is a lightweight code editor for a wide variety of operating systems [29].
- An emulator that works on a blockchain can be used to run a wide range of kinds of tests and commands by utilizing Ganache, a blockchain-assisted emulator. In order to run tests, deploy apps, and establish contracts, you can use a personal Ethereum blockchain called Ganache that you can access throughout the browser [30].
- A browser extension known as Metamask is used to connect to dispersed web pages by connecting to the Internet. Rather than running the complete Ethereum node in the browser, it runs Ethereum decentralized apps that are run in the browser [31].
- A hash string path can be used to transfer files using the distributed open storage system IPFS. It is employed to keep protected data that includes other data. The pathways work in a manner comparable to the traditional web URI. As a result, using their hash, all IoT data can be viewed at any time.

### 4.2. Proposed Solution—Algorithms

The Algorithms' Interpretation: the computational stages, data storage operations, and algorithm flow. By mapping the processes with algorithmic steps, the consistency between the proposed solution (Figure 5) and algorithmic specifications (Algorithm 1) is preserved.

---

**Algorithm 1** Contract Creating

---

1: Input: $\sigma$, L, $h(\gamma\wp)$, $\Delta p$, $\psi$, $\rho D$, $\Phi p$ Sensor, Location, Hash, Description
2: Uploading Type, Date, Blockchain Address
3: Output: bool
4: **procedure** SMARTCONTRACT
5: **if** msg.sender is not $\Phi p$ **then** Get Blockchain address to execute the smart contract
6: throw;
7: **end if**
8: mapping $h(\gamma\wp)$ to ($\sigma$ / L / $\rho D$) Map with each parameter
9: **end procedure**

---

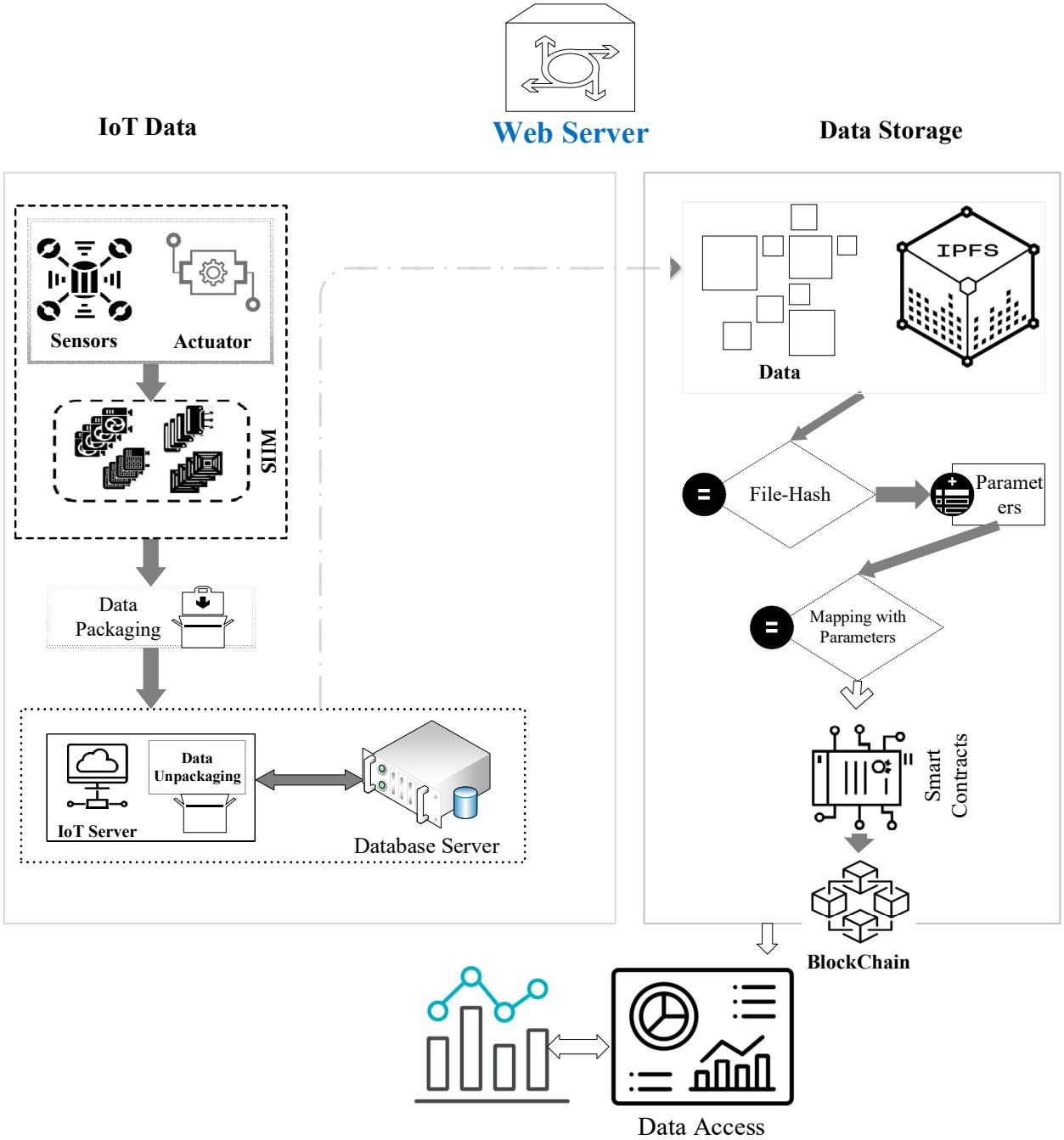

**Figure 5.** Overview of the detailed solution.

The functionality for uploading data is described in this section and seen in Algorithm 2. The technique is used to save the file hash and mapping of some other attributes in a smart contract and upload the data to IPFS. With a hash of the uploaded data, various parameters are mapped (date, uploading type, location, sensor, and description).

- Input(s): The input to the algorithm is used to map the parameters with a hash key.
- Processing: IoT data is read from the file and converted into a buffer, which is then uploaded to IPFS as an IoT data file and gives the hash key. A smart contract is used to record the uploaded data's hash key in the blockchain along with the extra attributes like sensor, location, description, uploading type, and date.
- Output: To save the mapped data in the blockchain is the result.

---
**Algorithm 2** Uploading Data

---
1: Input: $\sigma$, L, $\Delta p$, $\psi$, $\rho D$, $\Theta \lambda$ Sensor, Location, Description
2: Uploading Type, Date, Meta Data File
3: Output: *R* Uploading Message
4: **procedure** DATAUPLOADINGMODULE Event based function
5: **if** $\psi$ == User || $\psi$ == System **then** Uploading by User OR System
6: FS ← File($\Theta\lambda$) Get File stream
7: FB ← *Buffer.form* (FS) Convert to Buffer
8: FH ← *IPFS.Add* (FB) Get Hash of Uploaded Data
9: R ← SBC($\sigma$, L,FH, $\Delta p$, $\psi$, $\rho D$) Store Data to Blockchain with file hash
10: **end if**
11: **end procedure**

---

The data accessing functionality is validated in Algorithm 3 and specified in this section. Data from the blockchain is accessed using the protocol and made publicly accessible to users. In accordance with the necessary criteria, the user can obtain the data from the blockchain. There are different types for accessing the data. A user can access the data based on sensor, location data, and sensor with location mapping.

- Input(s): The parameters used to obtain the data are mapped using the algorithm's input.
- Processing: The data could be accessed from the blockchain based on different selections such as sensor, location, date, and sensor mapping with a location.
- Output: The output is available mapped data to public access.

---
**Algorithm 3** Data Access

---
1: Input: $\sigma$,L,$\rho D$ Sensor, Location, Date
2: Output: *R*, $\mu$
3: **procedure** INTERFACEMODULE
4: **if** $\sigma$ == N **then**
5: $\mu$ ← *GetData*($\sigma$) Get Data against Sensor
$\rho D$ == N
6: $\mu$ ← *GetData*($\rho D$) Get Data against given Date
L == N
7: $\mu$ ← *GetData*(L) Get Data against Location
$\sigma$ == N && L == N
8: $\mu$ ← *GetData*($\sigma$,L) Get Data against Sensor Location
9: **end if**
10: R ← UpdateDashboard($\mu$) Update available data on user screen
11: **end procedure**

---

The platform where all scenarios are successfully executed is shown in Figure 7 of the case study we are about to give, which includes the developed algorithms. Figure 6 shows how stakeholders submit the IoT dataset to IPFS's decentralized storage, and that data is then published to IPFS. The from date, to date, list of sensors, and list of locations where all the sensors are deployed are some of the custom parameters used by the custom query. Figure 7 depicts the internal blockchain ledger where we store the IPFS dataset, uploading information together with the dataset hash. Several sensors have been deployed in the area, and some of these include temperature, salinity, and pH sensors.

Sensors List

Select...

Available Locations

Select...

Upload IoT Data Package    Choose file

Browse

UPLOAD DATASET

**Figure 6.** Case Study Trail Performed.

CONTRACT
DatasetsStoring

ADDRESS
0×c9Ef8B8B5339Be517392d5034185B3dfD89f72A1

FUNCTION
AddCustomDataset(_name: string, _loc: string, _sensor: string, _sdate: string, _edate: string, _fhash: string)

INPUTS
My Custome Dataset, All-Locations, All-Sensors, , , QmNb8B8t15DAkyVhVk1gXj9UyfEK8131o6ELhFWrCrX6rZ

**Figure 7.** Data in the blockchain ledger.

*4.3. Algorithmic Execution of Tools and Technologies*

This section summarizes how relevant technologies and tools complement the suggested solution. In this debate, readers are encouraged to gain a better understanding of technology in general. A stack of technologies and tools is depicted in Figure 8. For instance, the sensor data is put into a CSV file and then encrypted and posted to the IPFS network, producing a hash key. The NodeJS framework has several tools that are utilized to generate a server-side application. We used VSC to start the NodeJS application. To rapidly build a personal Ethereum blockchain that you can use to run tests, issue commands, and examine the state while controlling how the chain functions, we used the Ganache Truffle Suite package.

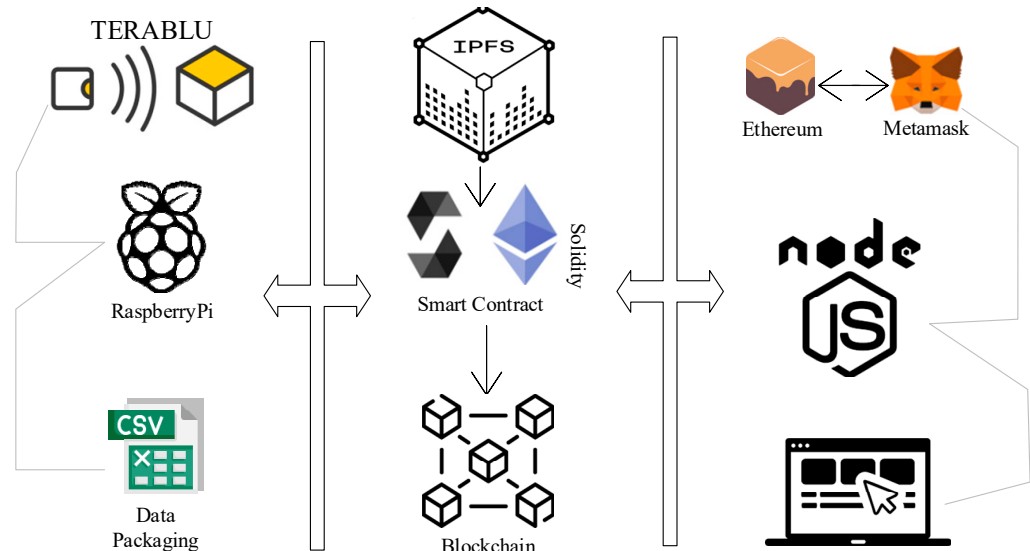

**Figure 8.** Overview of System Implementation Tools and Technologies.

**5. Evolutions and Validity Threats**

This section presents the results of the proposed solution. The evaluation setting is examined first, and then the operation of smart contracts as measured by fuel consumption. Then, using metrics like efficiency, we gauge and assess data uploading and storage to

the blockchain as well as query answer, including performance and algorithmic execution. Using the ISO/IEC-9126 model as the basis for the assessment criteria [32], In software-intensive systems, it is often necessary to use a quality evaluation tool to assess their performance. Additionally, a risk of the validity of this study is discussed, as well as possible limitations that must be taken into account in future research.

*5.1. Evaluation Environment*

Hardware and Software

A collection of hardware and software resources is used in the evaluation environment in order to run the solution, which can also be used to keep track of every step of execution and the result of the solution. Evaluation tests were conducted on the hardware side using both manual user input and automatic IoT data uploading via the Windows Platform (core i7 with 16 GB of runtime memory). Through execution evaluation, also referred to as evaluation scripts in the world of software, system testing is automated. Similar NodeJS scripts written in the ReactJS programming language were executed in Visual Studio Code. Additionally, the review process makes use of a variety of already-existing libraries, containing but not limited to ipfs.http, web3, and react. Using a JavaScript performance library script, for example, the CPU consumption of data is monitored when data is being uploaded to IPFS and placed on a blockchain, as well as when it is being retrieved from the blockchain using a JavaScript performance library script. To create a local Ethereum blockchain environment, a Ganache suit is employed, and a browser extension called Metamask is used to enable connections to distributed websites. In order to make use of gas transaction fees for the purpose of carrying out system functions, the Ganache suit and Metamask extension are linked to local Ethereum accounts.

Without paying for gas, the Ethereum smart contract cannot be carried out. In order to compare the fuel needed for the two methods of uploading the data, the fuel utilized to upload the original data was measured. The smallest unit of Ethereum money, the Gwei, is used to quantify fuel consumption. $10^9$ Wei is referred to as Gwei.

The price of contract migration execution is indicated in our proposed system (see Table 2). The price is specified in ether and includes the gas utilized. The amount of gas consumed multiplied by the price of gas equals one unit of ether. In this arrangement, the gas spent stands in for the continuous cost of computing. The value variations of ether in the account, the network has changed the price of gas [33].

**Table 2.** Analyse of the costs associated with data storage.

| Execution Type | Gas Used | Cost in Ether |
|---|---|---|
| Contract Creation | 2,027,188 | 0.04054376 |
| Contract Migration Call | 27,363 | 0.0054726 |
| Initial Contract | 225,237 | 0.0450474 |
| Initial Migration Call | 42,363 | 0.0084726 |

In the working prototype of our system, we automatically establish a gas restriction. The cost of creating the contract is 0.04054376 in ether, and the total amount of gas utilized is 2,027,188. The migration requires the establishment of contracts, which has a relatively low cost of 0.0027363 (ether) and uses only 27,363 of gas. If the input data is little in size, the general costs can be decreased even more.

The duration of time required for users to share data with others was the final test item. Data sharing time is a measurement of the overall amount of time spent reading, recalling, and sharing data. The outcomes of several sets of trials we ran with an average data size are displayed in Figure 9. A 450-byte upload consumes an average of 671,807 gas; a 1500-byte storage consumes an average of 1,942,901 gas. The data size increases with fuel

consumption. When IoT data was transferred to IPFS using the suggested system, there was no discernible difference in fuel consumption despite the increased quantity of data.

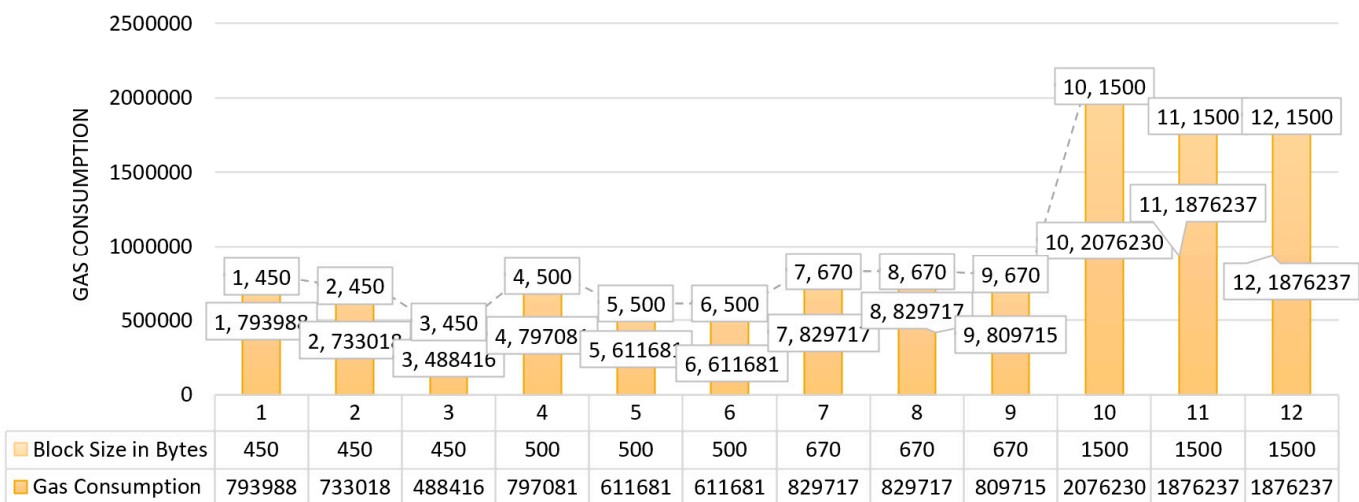

**Figure 9.** Gas consumption is based on transaction count and block size.

All of the network's entities and their interactions are depicted in a sequence diagram in Figure 10. Five distinct entities exist. Only the administrator, who has direct access to the dataset, uses the manual uploading entity. The dataset-based time cycle in the system is uploaded using a system uploading entity. Any stakeholder with public access might be a user entity and could access the data using their own custom queries. For the purpose of illustrating how the system works, Figure 9 depicts the execution flow.

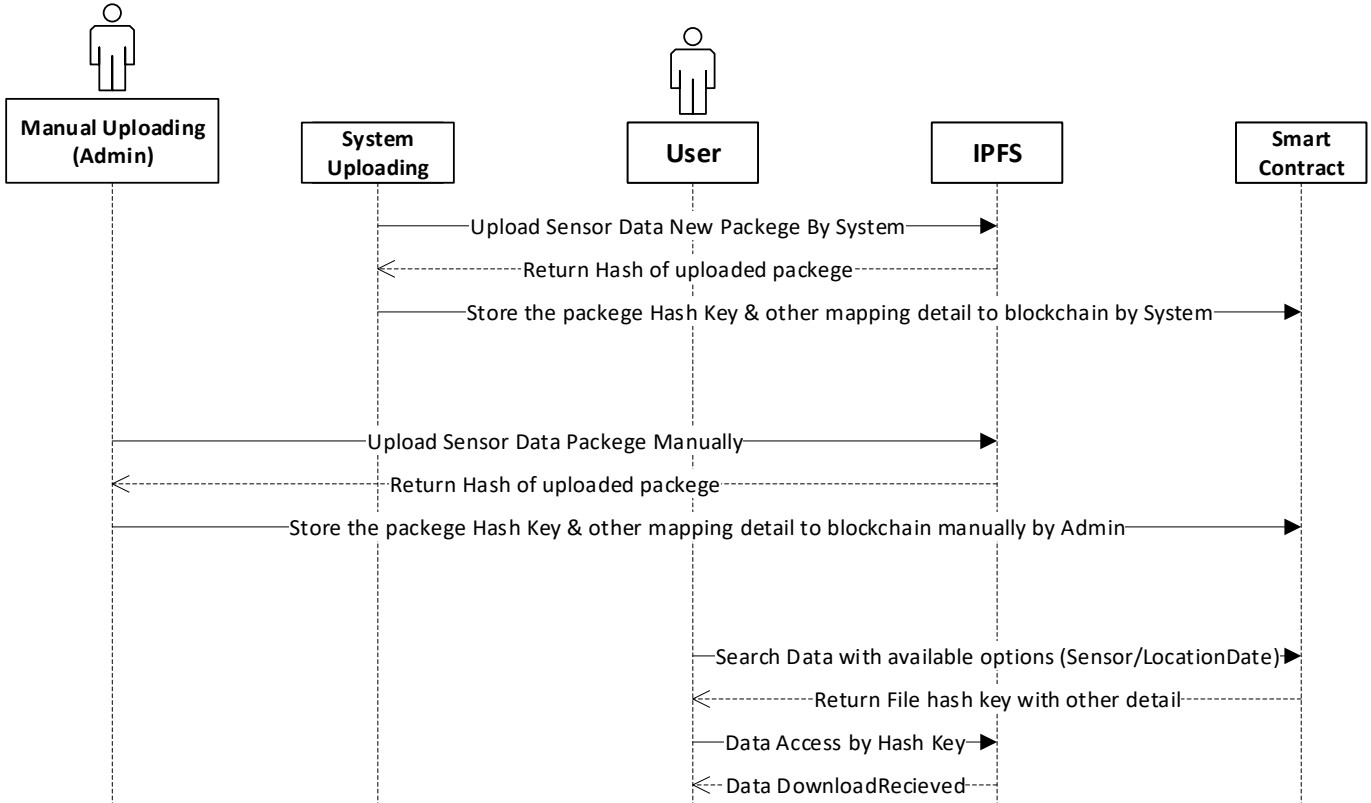

**Figure 10.** Data sharing process.

### 5.2. Evaluations of Query Response Time

Data querying is the key entity needed to store IPFS data packages and chain the details of IoT records. The performance of the solution in terms of storing and retrieving data from the blockchain may be evaluated using the query response time. Test results were conducted on two different methods: IPFS for IoT data storage and blockchain for file hashes. According to Figure 11, the horizontal axis indicates the two different execution functions, while the vertical axis is the response time as measured in milliseconds. As you can see from the title "Complete function", it explains how the entire method will be implemented from the moment the IoT data package is stored in IPFS to the point the record details are saved to the blockchain using the file hash that was created. "Smart Contract Function" shows the delay because of the Smart Contract execution call through Metamask. During the execution of a collection of functions using smart contracts, we also evaluate the performance of CPU consumption (see Figure 12). As with the data exchange, we assessed each stage of every strategy. There are several approaches, including calling encryption of the dataset, storing data in the blockchain ledger, and uploading it to IPFS storage.

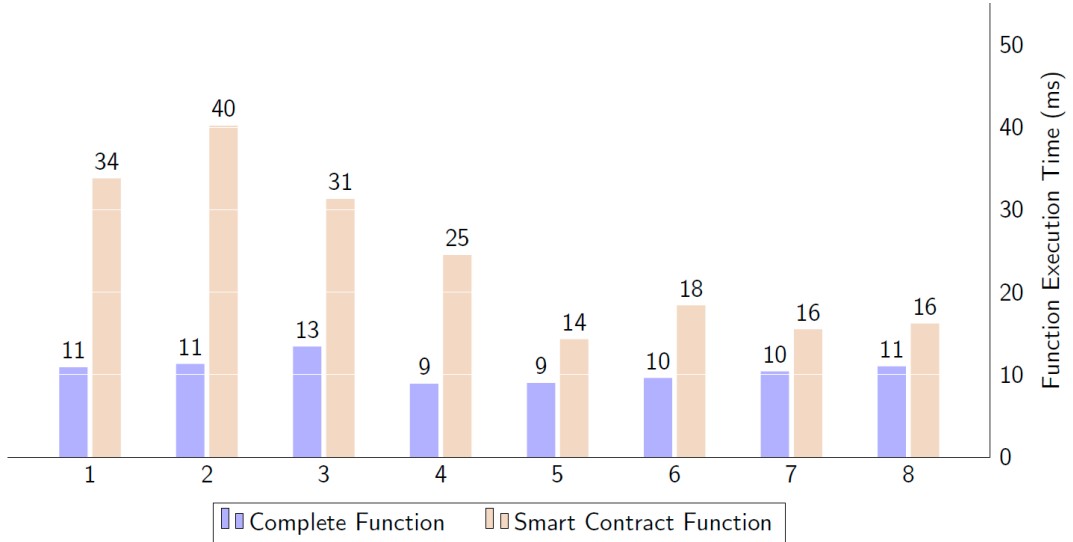

**Figure 11.** The time required to execute a function and keep the data in IPFS and Blockchain.

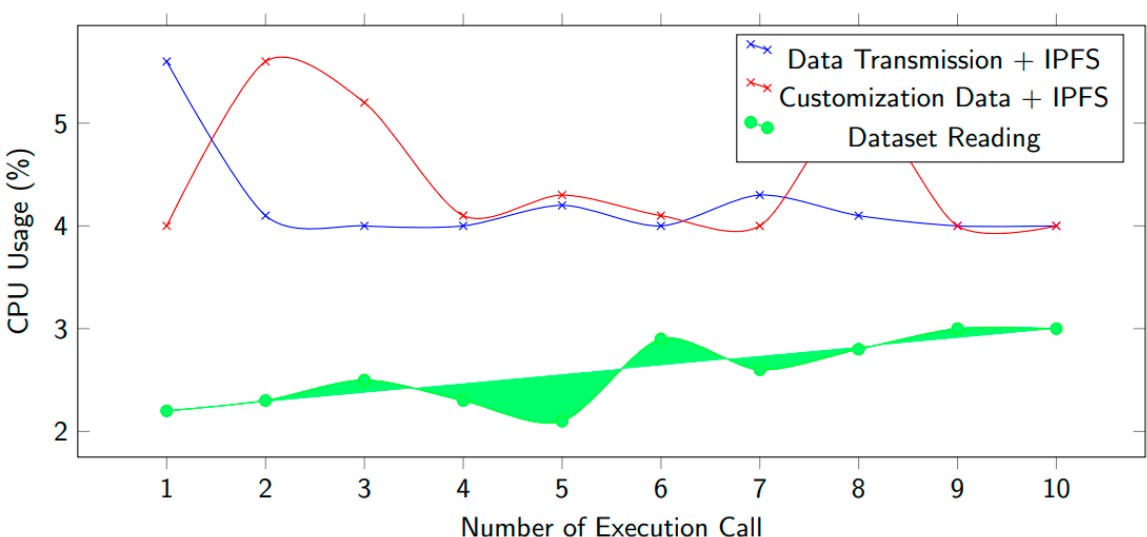

**Figure 12.** CPU time spent performing calculations.

## 6. Conclusions and Future Work

Typically, Internet of Things systems refer to a group of pervasive systems that take advantage of embedded sensors, applications, and networks in order to provide intelligent surroundings and systems. In order to make IoT data sharing and storage effective, it is crucial that a framework is put in place that enables IoT data storage in impromptu, unsafe settings. During the development of a reliable and distributed access control system, we looked into blockchain technology, specifically Ethereum smart contracts, which can be used for sharing data from IoT devices. To provide a distributed and reliable access control mechanism, we used Ethereum smart contracts to share IoT data in a distributed and secure manner. The solution described in this article combines IPFS and the Ethereum blockchain to store IoT data securely. Users may save and manage access roles for their IoT data more easily with the use of smart contracts. The suggested workaround logs the hash value along with other information in a blockchain ledger and encrypts IoT data provided to IPFS' decentralized storage. As part of this experiment, data lengths of different sizes were used to assess the performance of data uploading and access. It was found that the higher the size of the data, the more efficient and faster the process of uploading can be achieved. The researchers have further established that the upload technique that uses the system costs the same amount of gas regardless of the size of the data when it comes to the consumption of fuel, even when the data size increases.

**Author Contributions:** Conceptualization, A.R. and A.A.; methodology, A.R., S.A.K.G. and A.B.A.; resources, W.K. and M.A. writing—original draft preparation, A.R.; writing—review and editing, A.R., M.A., A.A. and W.K.; Supervision, A.B.A.; funding acquisition, A.B.A. All authors have read and agreed to the published version of the manuscript.

**Funding:** This research has been funded by Scientific Research Deanship at University of Ha'il-Saudi Arabia through project number RG-22020.

**Institutional Review Board Statement:** Not applicable.

**Informed Consent Statement:** Not applicable.

**Data Availability Statement:** Not applicable.

**Conflicts of Interest:** The authors declare no conflict of interest.

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
