# Peer review of "IoT Data Sharing Platform in Web 3.0 Using Blockchain Technology"

_electronics, doi:10.3390/electronics12051233_

Round 1

Reviewer 1 Report

The authors have presented a well written manuscript except for some spelling mistakes, like IoTT mentioned twice in the document. Including the practical approach rather than just the theory made the paper more sound.

  1. The abstract talks about the lack of trust in IPFS devices, and the paper is utilizing the blockchain algorithm to introduce the concept of trust into IPFS devices. The paper is well presented, with minor spell checks and minor adjustments that are needed for figures. Along with the theory of the said issue the authors have also shown a practical implementation.
  2. The conclusion and the results of the paper are in line with the abstract as well as the paper.
  3. Figure 1 seems to be very generalized and can definitely be improved by having information related to paper present in the figure.
  4. Figure 3 has a scope of improvement as well. I understand the authors idea, but the figure doesn’t fully capture the proposed solution.
  5. The references used are consistent with the contents in the paper.

Author Response

Please see comments as per reviewer number.

Reviewer 2 Report

The paper describes a system for securely storing IoT data. The solution combines the IPFS database and smart contracts to store data on the Ethereum Blockchain for secure data storage. The authors also evaluated data upload performance and fuel consumption accounting of these operations, which is valuable information.

In lines 60 and 128, the abbreviation IIOT needs to have the long form presented.

This paragraph needs better writing: Whether purposefully or accidentally, the corresponding recipient who released the data should be found and held accountable if a data leakage incident occurs (for instance, the sender has the information of the data breach and obtains the leaked data via the Internet).

Figure 4 should show two data flows, the data upload, and the opposite flow, with the return of the IPFS hash code.

The authors store the hash code of the IPFS, which is used as a key to retrieve its data. Each blockchain block has a hash code, which is used as a key to retrieve the block contents. The authors must explain if they also store the blockchain hash code to retrieve the data from the blockchain. 

For a better understanding, the authors should use flowcharts and data flow diagrams instead pseudocodes.

Figure 5 shows an IoT Server and a Database Server. The authors must explain which IOT Server and Database Server they used.

The authors mention Metamask for the Solution Implementation, but it did not appear in any diagram.

The authors did not mention which sensors they used and which data the sensors provided for this test, e.g., temperature, humidity, battery voltage, etc.

Author Response

(The authors gave the same response as above.)

Reviewer 3 Report

1- the contributions are not clear in abstract and introduction 

2- novelty of methodology is explained well

3- figure 1 need more explanation 

4- related work should be updated with more recent work such as Blockchain-empowered security and energy efficiency of drone swarm consensus for environment exploration, Blockchain-empowered digital twins collaboration: smart transportation use case, Drones’ edge intelligence over smart environments in B5G: Blockchain and federated learning synergy

5- the process of data sharing is not clear 

Author Response

Please see the response as per reviewer number.

Reviewer 4 Report

Dear author,

The article is suficient but i see right away something that for me is very importante

The lack of discussion section. The related works section is still poor.

Best regards 

Author Response

Please see the response as per the reviewer number.

Reviewer 5 Report

Internet of Things (IoT) driven systems can sense context-sensitive data but are prone to some unprecedented of data security and systems scalability challenges in an era of data-driven intelligence. State of the art in IoT-driven data sensing and sharing relies on third-party source of information (TTP) that accumulates data from one party and transmit to the other. Consequently, such IoT systems suffer from issues including but not limited to security, transparency, trust, and immutability as a result of TTP's involvement. Moreover, a multitude of technical impediments such as computation and storage poverty of IoTs, privacy concern, and energy efficiency enhances the challenges for IoTs.

The authors to address these issues of IoT security,  proposed a block-chain-enabled open IoT data-sharing framework based on the potential of the interplanetary file system (IPFS).

They used a case study-based approach to evaluate the proposed solution. The proposed scenario is implemented by building smart contracts in Solidity and deploying them on the local Ethereum test network.

They concluded stating that by incorporating the access roles in IoT data sensing as smart contracts of the blockchain, the proposed solution advocates for blockchain-oriented data security for IoT systems.

The study is interesting. However it need some improvements before being accepted.

These are my comments:

1.       The abstract must better summarize the sections.

2.       The components in figure 1 must be described in details. Also check this in the other figures.

3.       It is ok to add the paper organization in par. 1.1. I also suggest to add a par. with a clear purpose.

4.       There are two routines from row 224 up to row 239 directly in the text. I suggest, if they are important to put them inside to two boxes or in an appendix

5.       Avoid short par. (see for example par. 4.1.3)

6.       Par. 4.2 is hard to follow. The tables and the figures must be improved, better clarified and described in details.

7.       Add the discussion with the comparison to the literature and the limitation of the study

Author Response

(The authors gave the same response as above.)

Reviewer 6 Report

IoT Data Sharing Platform in Web 3.0 using Blockchain Technology is presented in this work which has provided the integration of three prominent platforms and technologies together. This work is well presented however the following minor issues can be addressed.

The suggested model presented in the introduction part can be moved to later chapters.

Proposed algorithms are presented like images that should be changed to text.

In which system environment the execution time is calculated?

Is any similar model proposed in the past by integrating all these?

if so some comparison of this model with the other existing model should be conducted and that should also be presented.

Author Response

Please see the response as per the reviewer's number.

Round 2

Reviewer 3 Report

Authors address the comments and orgnization and presentation of paper need to be improved 

Reviewer 4 Report

Accepted after minor revisional.

Best regards.

Reviewer 5 Report

N/A